# Phylogenetic and evolutionary analysis of dengue virus serotypes circulating at the Colombian–Venezuelan border during 2015–2016 and 2018–2019

Marlen Yelitza Carrillo-Hernandez[1,2], Julian Ruiz-Saenz[1], Lucy Jaimes-Villamizar[3], Sara Maria Robledo-Restrepo[2], Marlen Martinez-Gutierrez[1] *

**1** Grupo de Investigación en Ciencias Animales-GRICA, Universidad Cooperativa de Colombia, Bucaramanga, Colombia, **2** Programa de Estudio y Control de Enfermedades Tropicales-PECET, Universidad de Antioquia, Medellín, Colombia, **3** Laboratorio Clínico, E.S.E. Hospital Jorge Cristo Sahium, Norte de Santander, Colombia

\* marlen.martinezg@campusucc.edu.co

## Abstract

Dengue is an endemic disease in Colombia. Norte de Santander is a region on the border of Colombia and Venezuela and has reported the co-circulation and simultaneous co-infection of different serotypes of the dengue virus (DENV). This study aimed to conduct a phylogenetic analysis on the origin and genetic diversity of DENV strains circulating in this bordering region. Serum samples were collected from patients who were clinically diagnosed with febrile syndrome associated with dengue during two periods. These samples were tested for DENV and serotyping was performed using reverse transcriptase-polymerase chain reaction. Subsequently, positive samples were amplified and the envelope protein gene of DENV was sequenced. Phylogenetic and phylogeographic analyses were performed using the sequences obtained. Basic local alignment search tool analysis confirmed that six and eight sequences belonged to DENV-1 and DENV-2, respectively. The phylogenetic analysis of DENV-1 showed that the sequences belonged to genotype V and clade I; they formed two groups: in the first group, two sequences showed a close phylogenetic relationship with strains from Ecuador and Panama, whereas the other four sequences were grouped with strains from Venezuela and Colombia. In the case of DENV-2, the analysis revealed that the sequences belonged to the Asian–American genotype and clade III. Furthermore, they formed two groups; in the first group, three sequences were grouped with strains from Colombia and Venezuela, whereas the other five were grouped with strains from Venezuela, Colombia and Honduras. This phylogenetic analysis suggests that the geographical proximity between Colombia and Venezuela is favourable for the export and import of different strains among serotypes or clades of the same DENV serotype, which could favour the spread of new outbreaks caused by new strains or genetic variants of this arbovirus. Therefore, this information highlights the importance of monitoring the transmission of DENV at border regions.

**Data Availability Statement:** All relevant data are within the manuscript and its Supporting Information files.

**Funding:** This work was funded by CONADI of the Universidad Cooperativa de Colombia (Grant 1882–2017) and the Ministerio de Ciencia Tecnología e Innovación that provided to Marlen Carrilllo Hernandez a scholarship (Announcement 573-2016).

**Competing interests:** The authors have declared that no competing interests exist.

## Introduction

Dengue is the most important vector-borne viral infection in tropical and subtropical countries, globally infecting between 50 and 100 million individuals per year; its incidence has increased 30 times in the last 50 years [1]. This emerging infectious disease affects people of any age, sex and socioeconomic status; however, the highest number of cases has been reported in America, where all the four dengue virus serotypes are known to circulate (DENV-1, DENV-2, DENV-3 and DENV-4).

The genome of the dengue virus (DENV) is a single-stranded RNA of positive polarity that codes for three structural proteins [capsid (C), membrane (M) and envelope (E)] and seven non-structural proteins [NS1, NS2A, NS2B, NS3, NS4A, NS4B and NS5] with considerable genetic variation as a result of the mutations caused by RNA polymerase [2] and intra-serotype recombination [3]. Moreover, the analysis of the variation in the E gene has revealed that each serotype has genetic variants that constitute phylogenetic groups called genotypes and intra-genotype lineages that differ according to geographical distribution [4]. Therefore, the molecular characterisation and phylogenetic reconstruction of these viruses circulating in a specific population can facilitate the understanding of the epidemiological patterns and effects of human migration on the spread of the virus [5].

Colombia is located in the north western region of South America and has highest incidence rate of dengue in the Andean region, including countries such as Bolivia, Ecuador, Peru and Venezuela [6]. DENV was first detected in 1971 in Colombia; DENV-2 caused the first outbreak in the Atlantic coast [7]. Subsequently, the presence of DENV-3 was initially recorded in 1975 [7]; however, it reappeared in 2001 [8], whereas DENV-1 and DENV-4 were detected in 1978 and 1982, respectively [7]. Hence, although the four serotypes occur on a recurrent basis in Colombia, DENV-2 has been involved in the most important outbreaks during the last 20 years [9]. Furthermore, as in Colombia, Venezuela has been classified as hyperendemic for dengue. The first known dengue outbreak occurs in 1964 by DENV-3 [10]; subsequently, in 1969 it was reported the presence of DENV-2, DENV-1 in 1978 [10,11] and DENV-4 in 1985 [11]. After 32 years of absence, DENV-3 reappeared in Venezuela in 2000 [10].

In Colombia, the presence of the disease in recent years has become more intense; approximately 25 million individuals in urban areas are at risk of contracting it [12]. The departments with the highest annual incidence rates of dengue are Valle del Cauca, Santander, Antioquia, Huila, Tolima and Norte de Santander [13]. In addition, Norte de Santander is the main border department with Venezuela, and it is recognized as the most populous and vibrant border [14] because it is located at the main land route connecting Colombia with Venezuela via the Simón Bolívar International Bridge (located in the municipality of Villa del Rosario). In this area, co-circulation and simultaneous arboviral co-infections (zika, chikungunya and dengue) have been reported, and different serotypes of DENV have been identified in the same patient cohort [15]; however, a robust phylogenetic analysis has not been conducted. Thus, this study aimed to perform a thorough phylogenetic analysis to identify the origin, evolution and genetic diversity of DENV-1 and DENV-2 strains that have been prevalent at the north-eastern border of Colombia in the last five years.

## Materials and methods

### Ethical considerations

This study was approved by the Bioethics Committee of the Universidad Cooperativa de Colombia (Act 0030635 of April 16, 2015 and Act 001 of September 5, 2018) and the scientific

committee of the ESE Jorge Cristo Sahium Hospital (Registration number #087, July 28, 2015 and #083, October 23, 2018). All individuals aged >18 years who met the clinical criteria provided written informed consent; parents or legal guardians provided written informed consent for children aged <18 years. The samples were anonymously analysed.

## Samples

The study samples were collected during two different periods from August 2015 to April 2016 [15] and from November 2018 to July 2019 from patients visiting ESE Jorge Cristo Sahium Hospital (ESEHJCS) located at Villa del Rosario. This municipality is located in the eastern region of department Norte de Santander, which is on the north-eastern border of Colombia and Venezuela (Fig 1A). Villa del Rosario is at an altitude of 1240 feet with an average temperature of 77.0˚F (25˚C) per year and a population size of 65,337 habitants. The study participants were patients of any age with a clinical diagnosis of febrile syndrome that was compatible with dengue; they were in the acute stage of the disease, i.e., fever for no more than seven days. The medical personnel of ESEHJCS performed clinical diagnosis based on the operational definition of probable dengue cases established in the protocol of public health surveillance of dengue in Colombia [16], which is based on the guidelines for the care of patients in the America's regions PAHO, 2015 [17]. A total of 157 and 72 serum samples were collected during the first and second periods, respectively.

## RNA extraction and reverse transcription

The QIAamp Viral RNA Mini Kit™ (QIAGEN © Hilden, Germany) was used to extract viral RNA according to the manufacturer's instructions. The quality and quantity of RNA was determined using a NanoDrop™ One Microvolume UV–visible spectrophotometer (Thermo Fisher Scientific®, Waltham, MA). cDNA was synthesised using RevertAid Reverse Transcriptase (Thermo Fisher Scientific®) according to the manufacturer's instructions with at least 0.5 μg RNA and random primers for reverse transcription.

## Reverse transcriptase–polymerase chain reaction (RT–PCR) and sequencing

For DENV identification, conventional PCR was performed using previously reported primers: mD1 and D2 [18]. Amplification was performed using the Maxima Hot Start Green PCR Master Mix (2×) (Thermo Scientific®) with 10 μMol per primer and 2 μl cDNA per sample. The amplification conditions included denaturation for 5 min at 95˚C, followed by 35 cycles at 95˚C for 30 s, 55˚C for 45 s and 72˚C for 33 s, with final extension at 72˚C for 10 min. The amplification was confirmed in 1.5% agarose gel, and the expected size of the PCR products was 511 bp. Subsequently, the positive samples were processed for serotype identification using multiplex RT–quantitative polymerase chain reaction (qPCR) technique as previously described [18].

For amplification of the partial E gene of DENV-1 and DENV-2, previously reported primers were used [19]. Conventional PCR amplification was performed using the Maxima Hot Start Green PCR Master Mix (2×) (Thermo Scientific®) with 10 μMol of each primer and 5 μl of the cDNA of each sample. The amplification conditions included denaturation for 5 min at 95˚C, followed by 35 cycles at 95˚C for 30 s, between 54˚C and 56˚C for 45 s (depending on the primers of each serotype) and 72˚C for 2 min, with final extension at 72˚C for 10 min. One-step RT-PCR was applied to samples that did not amplify previously. Briefly, SuperScript ™ III One-Step RT-PCR System with Platinum ™ Taq High Fidelity DNA Polymerase (Life Technologies™, Paisley, UK) was used, by adding 12.5 μL 2× reaction mix, 1 μL of the enzyme,

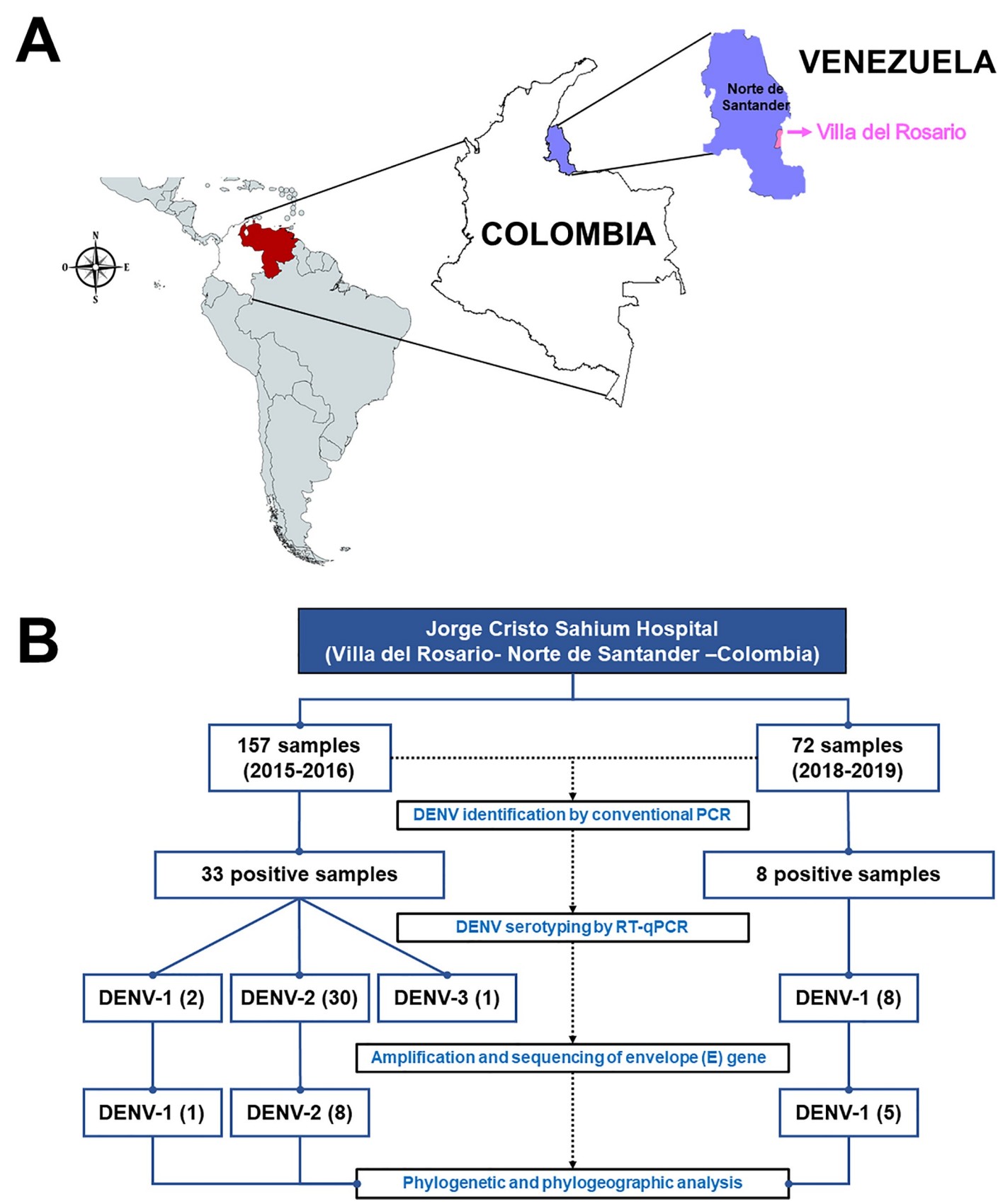

**Fig 1. Geographical origin of the sampling site in Colombia and the process of sample collection. (A).** The blue area indicates the Norte de Santander region, and the pink area indicates Villa del Rosario City. The map was created using DIVA-GIS software (V.7.5.0 for Windows™ and mapchart.net (https://mapchart.net/americas.html). **(B).** Process of sample collection. The samples were collected during two periods. The first period was between August 2015 and April 2016 and the second was between November 2018 and July 2019. For DENV identification, conventional PCR was performed; multiplex RT–qPCR was performed for serotyping. Finally, the envelope (E) gene was amplified and sequenced and the phylogenetic and phylogeographic analyses were conducted.

1 µL (10 µM) of previously reported primers for the partial E gene of DENV-1 and DENV-2 [19], 0.5 µL of the extracted RNA and nuclease-free water. The amplification conditions included 60˚C for 30 min, 94˚C for 2 min followed by 45 cycles of 94˚C for 15 s, 55˚C for 30 s, 68˚C for 2 min and a final extension of 68˚C for 5 min.

The expected sizes for each amplification were 1818 bp for DENV-1 and 1797 bp for DENV-2, which were visualised in 1% agarose gel using EZ-Vision® (AMRESCO, TX, USA) and the documentation gel system GEL DOC™ XR+ (BioRAD Hercules, CA, US). Finally, the amplified PCR products were purified and sequenced using Macrogen Inc. (Seoul, Korea) in an automated ABI 3730xl sequencer.

## Phylogenetic analysis

The sequences obtained were edited and assembled using SeqMan PRO (DNASTAR Inc. Software, Madison, WI, USA). Subsequently, they were confirmed using the basic local alignment search tool (BLAST). Next, datasets were constructed with sequences taken from the GenBank. Complete genome DENV-1 or DENV-2 sequences and complete or partial E gene sequences were downloaded, belonging mainly to genotype V and Asian–American, respectively (S1 and S2 Tables). In addition, these sequences included information on the country and sampling dates. This resulted in a final dataset of 131 sequences for DENV-1 and 142 sequences for DENV-2; subsequently, the sequences were aligned using the CLUSTAL W program [20], that generated an E gene sequence alignment for DENV-1 with a length of 945 nt and for DENV-2 with a length of 1178 nt. Afterward, phylogenetic trees were constructed using the maximum likelihood (ML) method from the MEGA software 7.0 [21], with a bootstrap value of 1000. Furthermore, the jModelTest tool was used [22] to select the most appropriate nucleotide replacement model, considering the Akaike (AIC) and Bayesian (BIC) information criteria. The strain KJ189304 served as an outgroup for DENV-1 and strain EF105379 served as an outgroup for DENV-2 phylogeny. Genetic distances were calculated using the p-distance replacement of nucleotides and amino acids model.

The mean nucleotide substitution rate per site per year (substitution/site/year), time to the most recent common ancestor (TMRCA), geographic origin and overall spatial dynamics were inferred using the Bayesian Markov chain Monte Carlo (MCMC) statistical framework implemented using the BEAUti/BEAST v1.8.4 package [23]. The analyses used for both serotypes were a relaxed uncorrelated lognormal molecular clock and reversible discrete phylogeography model. The best-fit demographic model executed for DENV-1 had a constant size and that for DENV-2 was an exponential model. For both viruses, 4E08 generations were run to ensure an effective population size of >200 for the evaluated parameters using the Tracer v1.7 program [24]. Maximum clade credibility (MCC) trees were summarized using TreeAnnotator v1.8 [23] and visualized using FigTree v1.4.2 [25].

## Results

The samples were collected from the Norte de Santander region during the periods of August 2015 to April 2016 and November 2018 to July 2019. Villa del Rosario is a municipality located in the department of Norte de Santander (Colombia) on the border of Venezuela; it is part of

**Table 1. Demographic information of patients from whom the DENV E protein genes were sequenced.**

| Sample code | Gender | Age (years) | Collection date | Serotype |
|---|---|---|---|---|
| 11 | Male | 13 | September 2015 | DENV-2 |
| 22 | Male | 18 | September 2015 | DENV-1 |
| 32 | Female | 9 | October 2015 | DENV-2 |
| 46 | Female | 7 | October 2015 | DENV-2 |
| 77 | Female | 10 | December 2015 | DENV-2 |
| 95 | Male | 13 | December 2015 | DENV-2 |
| 106 | Female | 34 | January 2016 | DENV-2 |
| 108 | Male | 6 | January 2016 | DENV-2 |
| 129 | Female | 22 | February 2016 | DENV-2 |
| 39 | Female | 9 | April 2019 | DENV-1 |
| 50 | Female | 12 | May 2019 | DENV-1 |
| 54 | Female | 4 | June 2019 | DENV-1 |
| 60 | Male | 6 | June 2019 | DENV-1 |
| 62 | Male | 15 | July 2019 | DENV-1 |

the urban area of the Metropolitan Area of Cucuta. It is at an altitude of 440 m above sea level and has a population of 80,433 inhabitants, as reported in 2011 (Fig 1A).

During the first period, 33 samples were positive for DENV, including two for DENV-1, 30 for DENV-2 and one for DENV-3 [15]. Among the samples collected during the second period, eight samples were positive for DENV-1. The mean age of the patients was 12.71 years (SD ± 7.87 years). The patients included eight (57.14%) women and six (42.85%) men (Table 1).

The quality and quantity of amplification products obtained from nine of the 33 samples during 2015–2016 were sufficient to sequence the partial E gene. Six of these samples were collected between August 2015 and December 2015 and the other three were collected between January 2016 and April 2016. In five samples obtained during 2018–2019, an amplification product was obtained in enough quantity and quality to sequence the E gene (Fig 1).

## Phylogenetic analysis

BLAST analysis confirmed that six and eight sequences belonged to DENV-1 and DENV-2, respectively. On the other hand, the analysis of the DENV-1 and DENV-2 datasets using the jModel Test determined that TN93 + G + I was the appropriate model for the analysed region. A small number of samples were amplified; this may be attributable to the fact that the amplifications were obtained directly from the sample and viral isolation was not performed which would have permitted obtaining higher concentrations of genomic material.

The phylogenetic analysis of DENV-1 using the ML methods produced a tree where the five genotypes (I–V) can be observed (Fig 2) with a nucleotide difference between 7% and 17.8%. As the analysis focused on genotype V, three clades (I–III) were formed, with bootstrap value of >75% and one nucleotide difference among clades of 3.3–5.5% (Table 2). Clade I comprised sequences from across the American continent, primarily from Venezuela, Colombia, Brazil, Peru, Ecuador, Bolivia, Argentina, Paraguay, United States, Mexico, Nicaragua, El Salvador, Honduras, Costa Rica and some countries of the Greater and Lesser Antilles. Clade II comprised sequences from some countries of the Greater and Lesser Antilles, whereas clade III comprised sequences mainly of Asian origin (Fig 2). Conversely, two subclades called I–VE (Venezuela) and I-BR (Brazil) were established in clade I. The first subclade comprised

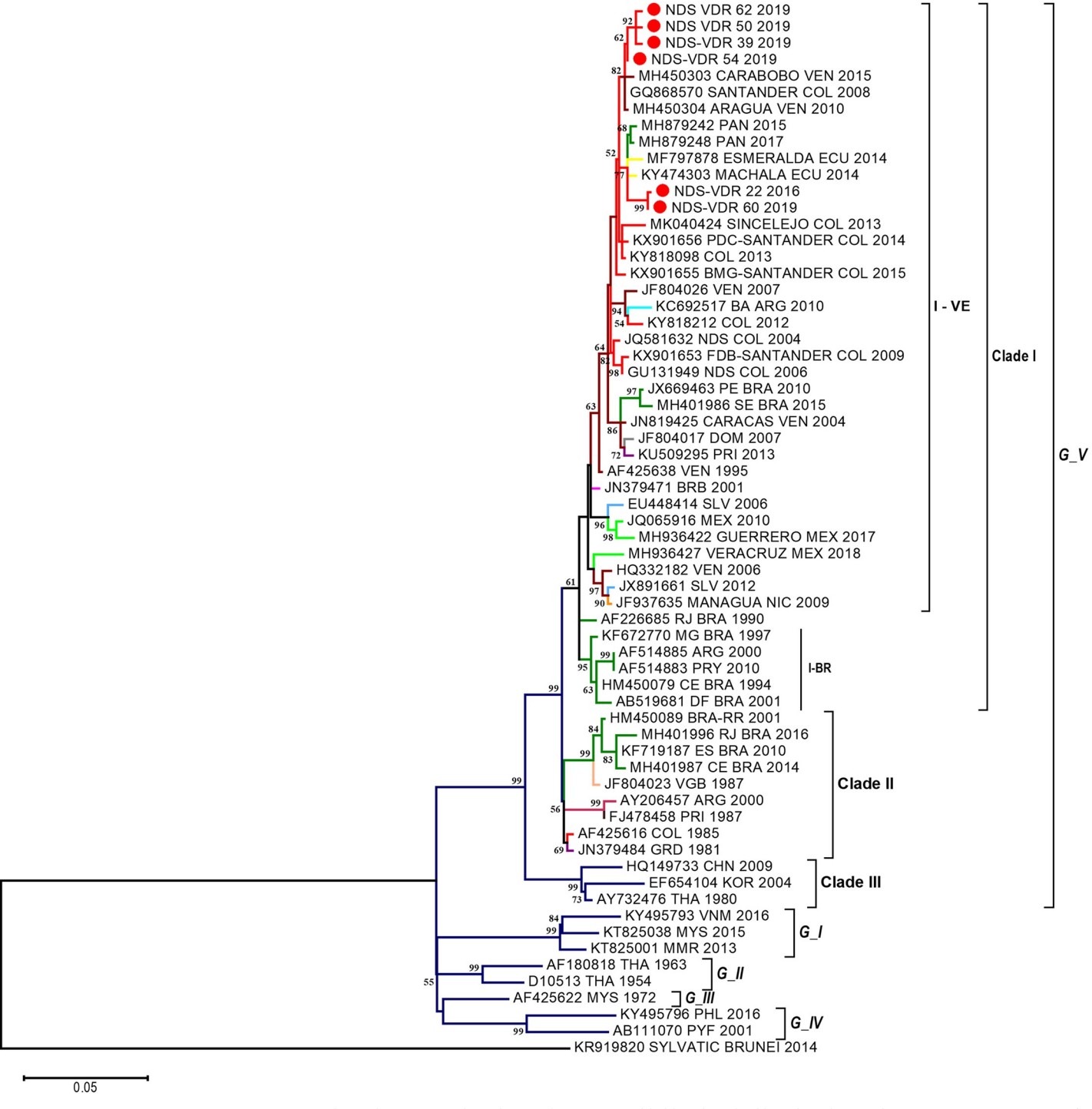

**Fig 2. Phylogenetic analysis of DENV-1 strains.** The analysis was conducted using the maximum likelihood method based on the Envelope gene using 6 sequences of 945 nt length and 131 sequences that have been previously deposited in GenBank. The tree was reconstructed using the nucleotide replacement model TN93 + G + I. The red circles represent the sequences from Norte de Santander identified as NDS-VDR. G: Genotype, BR: Brazil and VE: Venezuela. Color Code: Red: Colombia, red wine color: Venezuela, Blue: Argentina, Light Blue: Dominican Republic, green: Brazil, yellow: British Virgin Islands, and purple: Asian sequences.

**Table 2. Diversity of nucleotides in genotype V of DENV-1.**

| (A) | Gp_1 | Gp_2 | | | | |
|---|---|---|---|---|---|---|
| Gp_1 | — | 0.004 | | | | |
| Gp_2 | 0.019 | — | | | | |
| (B) | I-VE | I-BR | | | | |
| I-VE | — | 0.003 | | | | |
| I-BR | 0.027 | — | | | | |
| (C) | Clade_I | Clade_II | Clade_III | | | |
| Clade_I | — | 0.003 | 0.006 | | | |
| Clade_II | 0.033 | — | 0.006 | | | |
| Clade_III | 0.055 | 0.052 | — | | | |
| (D) | G_V | G_I | G_IV | G_II | G_III | Sylvatic |
| G_V | — | 0.007 | 0.007 | 0.007 | 0.007 | 0.010 |
| G_I | 0.089 | — | 0.007 | 0.007 | 0.008 | 0.010 |
| G_IV | 0.093 | 0.088 | — | 0.007 | 0.007 | 0.011 |
| G_II | 0.078 | 0.076 | 0.078 | — | 0.007 | 0.010 |
| G_III | 0.070 | 0.072 | 0.068 | 0.056 | — | 0.011 |
| Sylvatic | 0.178 | 0.184 | 0.180 | 0.173 | 0.170 | — |

Diversity within paraphyletic groups (A), subclades (B), clades (C) and among genotype V and others DENV-1 genotypes (D).

Paraphyletic group 1 (Gp_1): Sequences NDS_VDR 62, 50, 39 y 54.

Paraphyletic group 1 Gp_2: Sequences NDS_VDR 22 and 60.

I-VE: Subclade Venezuela.

I-BR: Subclade Brazil.

G_I to G_V: DENV-1 genotype I–V.

sequences from Central and South America, whereas the second subclade comprised sequences from Brazil with a nucleotide difference of 2.7% (Table 2).

The six Colombian sequences of this study formed paraphyletic grouping in genotype V, clade I and subclades I–VE. The sequences of this study were grouped together with sequences from other regions or countries and they all descend from a common ancestor, whereas a monophyletic grouping implies that only the sequences from this study were found grouping and descending from a common ancestor. Therefore, two sequences (one from 2016 and another from 2019) were grouped with strains from Ecuador (MF797878 and KY474303) from 2014, and Panama (MH879242 and MH879248) from 2015 and 2017, whereas the other four sequences obtained during 2018–2019 were grouped with a Colombian 2008 sequence (GQ868570) and two sequences from Venezuela (MH450304 and MH450303) from 2010 and 2015. The distance-p analysis revealed a nucleotide difference of 1.9% between these two sequence groups (Table 2).

In the case of DENV-2, a phylogenetic tree with six genotypes was obtained using the ML method (Fig 3), with a nucleotide difference between 7.2% and 25.7%. Moreover, three clades (I–III) were formed in the Asian–American genotype, presenting a nucleotide difference between 2.3% and 3.0% (Table 3).

Clade I comprised sequences from the Greater and Lesser Antilles, whereas Clade II comprised sequences from the Greater and Lesser Antilles and South America. However, clade III comprised sequences from South America, Central America and some countries in the Greater and Lesser Antilles. Moreover, four subclades were formed in this clade named South America (SA) I–IV and Central America (CA), with a nucleotide difference between 2.3% and 3.0% (Table 3).

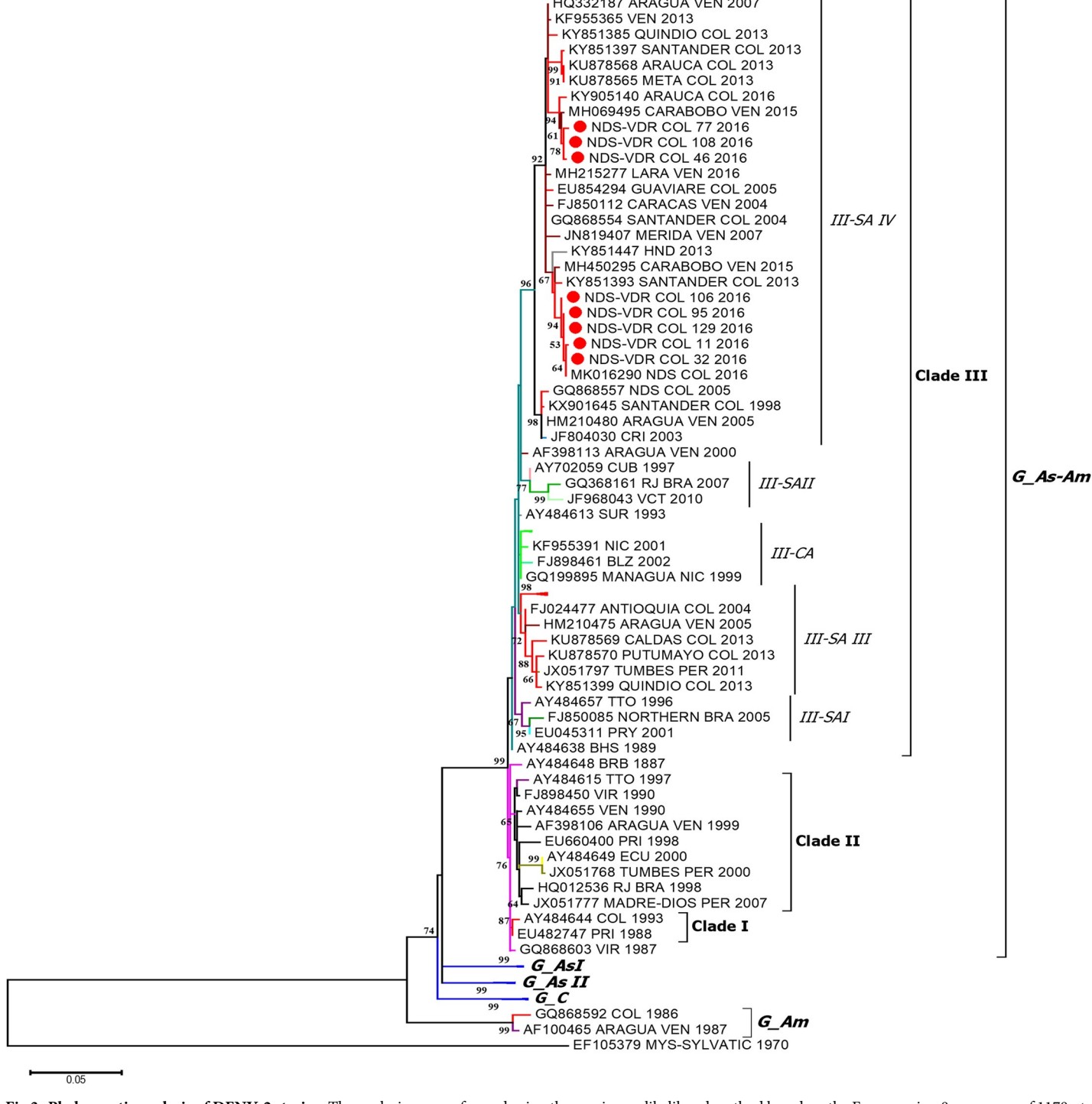

**Fig 3. Phylogenetic analysis of DENV-2 strains.** The analysis was performed using the maximum likelihood method based on the E gene, using 8 sequences of 1178 nt in length and 142 sequences that have been previously deposited in GenBank. The tree was reconstructed using the nucleotide replacement model TN93 + G + I. The red circles represent the sequences from Norte de Santander and identified as NDS-VDR. G: Genotype, C: Cosmopolitan, As: Asian, Am: American, SA: South America and CA: Central America. Color Code: Red: Colombia, Light Blue: Venezuela Light green: Mexico, Blue: Argentina, Green: Brazil, Orange: Ecuador, and purple: Asian sequences.

**Table 3. Diversity of nucleotides in Asian–American genotype of DENV-2.**

| (A) | Gp 1 | Gp 2 | | | | |
|---|---|---|---|---|---|---|
| Gp_1 | — | 0.005 | | | | |
| Gp_2 | 0.020 | — | | | | |
| **(B)** | **SA-IV** | **CA** | **SA-III** | **SA-I** | **SA-II** | |
| SA-IV | — | 0.004 | 0.004 | 0.004 | 0.005 | |
| CA | 0.022 | — | 0.003 | 0.003 | 0.004 | |
| SA-III | 0.030 | 0.016 | — | 0.003 | 0.004 | |
| SA-I | 0.028 | 0.014 | 0.021 | — | 0.004 | |
| SA-II | 0.032 | 0.020 | 0.027 | 0.026 | — | |
| **(C)** | **Clade_III** | **Clade_II** | **Clade_I** | | | |
| Clade_III | — | 0.004 | 0.003 | | | |
| Clade_II | 0.030 | — | 0.002 | | | |
| Clade_I | 0.023 | 0.016 | — | | | |
| **(D)** | **G_As/Am** | **G_Am** | **G_C** | **G_As_I** | **G _As_II** | **Sylvatic** |
| G_As/Am | — | 0.014 | 0.010 | 0.011 | 0.009 | 0.024 |
| G_Am | 0.114 | — | 0.013 | 0.014 | 0.012 | 0.024 |
| G_C | 0.082 | 0.102 | — | 0.01 | 0.008 | 0.025 |
| G_As_I | 0.085 | 0.115 | 0.080 | — | 0.010 | 0.024 |
| G_A II | 0.071 | 0.099 | 0.069 | 0.069 | — | 0.025 |
| Sylvatic | 0.257 | 0.248 | 0.265 | 0.251 | 0.264 | — |

Diversity within paraphyletic groups (A), subclades (B), clades (C) and among Asian–American and others DENV-2 genotypes (D).

Paraphyletic group 1 (Gp_1): Sequences NDS_VDR 77, 108 and 46.

Paraphyletic group 1 Gp_2: Sequences NDS_VDR 106, 95, 129, 11 and 32.

SA-I to SA-IV: Subclade South America I–IV.

CA: Subclade Central America.

G_As/Am: Asian–American genotype.

G_Am: American genotype.

G_C: Cosmopolitan genotype.

G_As_I to II: Asian genotype I–II.

It was also observed that the eight sequences collected during 2015 and 2016 formed paraphyletic grouping in the Asian–American genotype, clade III, and subclade SA-IV. Three of these sequences were grouped with those from Colombia (Arauca, Meta, Quindío and Santander; 2013–2016) and Venezuela (2007–2013), whereas the remaining five sequences were grouped with those from Colombia (Santander and Casanare; 2010–2016) and Venezuela (sequences from the year 2015). Furthermore, a nucleotide difference of 2% was identified using the p-distance analysis of these two sequence groups (Table 3).

## Phylogeographic analyses

The evolutionary rate was $5.44 \times 10^{-4}$ subs./site/year (95% HPD: $4.45$–$6.38 \times 10^{-4}$ subs./site/year) for DENV-1 and $7.37 \times 10^{-4}$ subs./site/year (95% HPD: $6.25$–$8.48 \times 10^{-4}$ subs./site/year) for DENV-2. The MCC tree for DENV-1 and DENV-2 evidenced multiple introductions intro the country, especially between Colombia and Venezuela.

The MCC tree for genotype V of DENV-1 included 68 sequences from Colombia and evidenced four introductions into the country (Fig 4). The first introduction included two sequences isolated in 1985 (AF425616) and 1996 (AF425617); and were related to viruses from

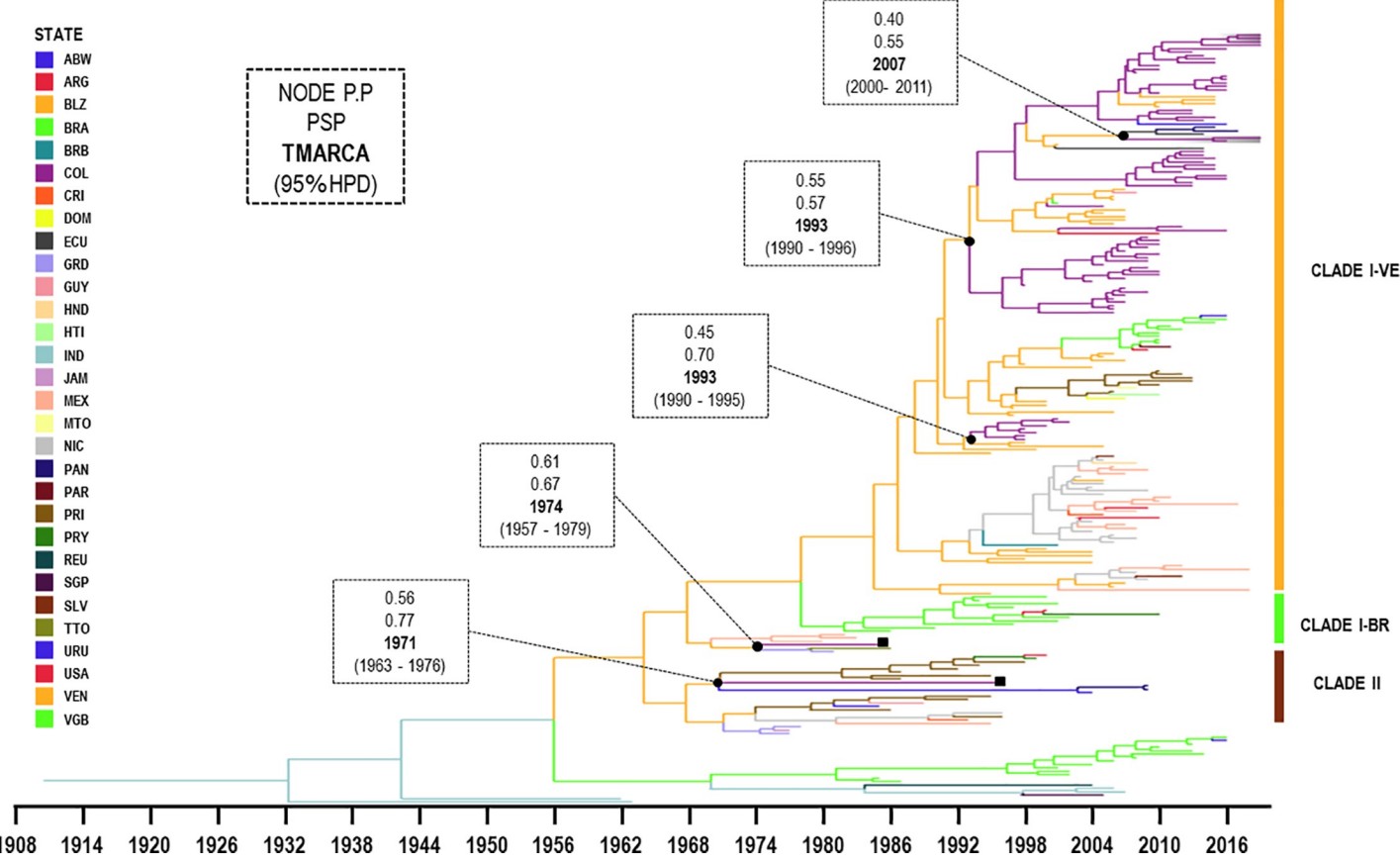

**Fig 4. Maximum clade credibility tree of DENV-1 corresponding to the genotype V.** Branches are coloured according to the most probable location of their parental node and legends shown on the left side. All horizontal branch lengths are drawn according to a scale of years. Two clades (I-II) and subclade of the clade I (VE: Venezuela, BR: Brazil) are identified on the right side and the gray frame represented our sequences isolated from Villa del Rosario, Norte de Santander. ■: virus strains isolated in 1985 (AF425616) and 1996 (AF425617) available in Genbank. PSP: posterior state probability, PP: posterior probability, TMARCA: the mean estimated time to the most recent common ancestor and 95% HPD: 95% highest posterior density interval. ABW: Aruba, CRI: Costa Rica, ECU: Ecuador, GRD: Grenada, COL: Colombia, GUY: Guyana, NIC: Nicaragua, MTQ: Martinique, MEX: Mexico, URU: Uruguay, PRY: Paraguay, PRI: Puerto Rico, JAM: Jamaica, PER: Peru, BRA: Brazil, TTO: Trinidad and Tobago, HTI: Haiti, VGB: British Virgin Islands, VEN: Venezuela, IND: India, SGP:Singapure and REU:Reunion.

Lesser Antilles and some countries of Central America, with a TMRCA of approximately 1971 (95% HPD: 1963–1976).

The second introduction included Colombian sequences isolated from the east of the country (Santander between 1998 and 2001) were grouped with sequences from Venezuela (1995–1999), with TMRCA of approximately 1993 (95% HPD: 1990–1995). Venezuela being the most probable ancestral location (posterior state probability [PSP] = 0.70 and posterior probability [PP] = 0.45).

The third introduction included sequences from Colombia (Santander and Norte de Santander; 2006–2010), Venezuela (1997–2006) and Brazil (2010–2016) with a TMRCA of approximately 1993 (95% HPD: 1990–1996).

Subsequently, frequent cross-border transmissions of DENV-1 between Venezuela and Colombia were observed. Cross-border transmission were also observed between Venezuela, Colombia, Ecuador and Panama with TMRCA of approximately in 2007 (95% HPD: 2000–2011).

The MCC tree for American–Asian genotype DENV-2 included 42 sequences from Colombia and four evident introductions into the country (Fig 5). In the first introduction, a single

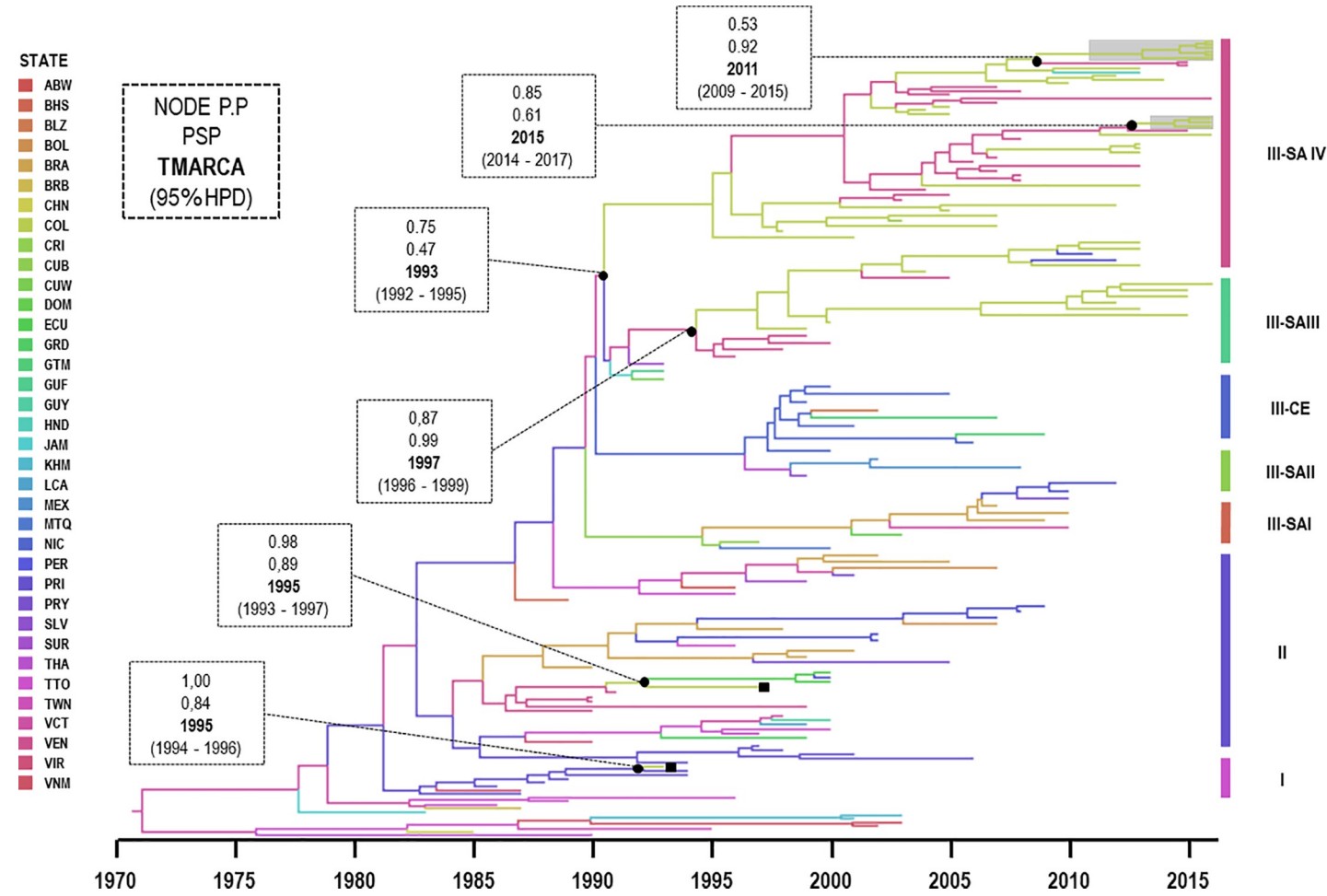

**Fig 5. Maximum clade credibility tree of DENV-2 corresponding to Asian–American genotype.** Branches are coloured according to the most probable location of their parental node and legend shown on the left side. All horizontal branch lengths are drawn to a scale of years. Three clades (I–III) and subclade of the clade III are identified on the right side and the gray frame represents our sequences isolated from Villa del Rosario, Norte de Santander. ■: virus strains isolated in 1993 (AY484644) and 1997 (DQ364497) available in Genbank. PSP: posterior state probability, PP: posterior probability, TMRCA: the mean estimated time to the most recent common ancestor and 95% HPD: 95% highest posterior density interval. ABW: Aruba, BHS: Bahamas, BLZ: Belize, BOL: Bolivia, BRB: Barbados, CRI: Costa Rica, CUB: Cuba, CUW: Curacao, DOM: Dominica, ECU: Ecuador, GRD: Grenada, GTM: Guatemala, GUY: Guyana, COL: Colombia, GUF: French Guiana, LCA: Santa Lucia, NIC: Nicaragua, MTQ: Martinique, MEX: Mexico, PRY: Paraguay, PRI: Puerto Rico, JAM: Jamaica, PER: Peru, BRA: Brazil, TTO: Trinidad and Tobago, HND: Honduras, SLV: Salvador, SUR: Surinam, VCT: Saint Vincent and the Grenadines,VIR: United States Virgin Islands, VEN: Venezuela, TWN:Taiwan, KHM:Cambodia, VNM: Vietman.

Colombian sequence was isolated in 1993 (AY484644) and was grouped with sequences from Puerto Rico (1987–1994). It had a TMRCA of approximately 1995 (95% HPD: 1994–1996) and Puerto Rico appeared as the most probable ancestral location (PSP = 0.84, PP = 1.00).

In the second introduction, another sequence from 1997 (DQ364497) was grouped with sequences from South American (Venezuela, Ecuador and Peru) and they had TMRCA of approximately 1995 (95% HPD: 1993–1997) and was most likely introduced from Venezuela (PSP = 0.89, PP = 0.98).

In the third introduction, other Colombian sequences belonging from the east of the country (Santander, Norte de Santander, Meta, Guaviare and Casanare between 2001 and 2012) were grouped with sequences from Venezuela (2003–2005), and Costa Rica (2003). These groups had a TMRCA of approximately 1993 (95% HPD: 1992–1995) and Saint Vincent and the Grenadines appeared to be the most probable ancestral location (PSP = 0.47, PP = 0.75).

In the fourth introduction, other Colombian sequences isolated from Santander, Tolima, Huila, Cundinamarca, Antioquia, Putumayo, Caldas and Quindío between 1999 and 2013 were grouped with sequences from Venezuela (1996–2005) and Peru (2011–2012). All of them had a TMRCA of approximately 1997 (95% HPD: 1996–1999) and were most likely introduced from Venezuela (PSP = 0.99, PP = 0.87).

Frequent cross-border transmissions of DENV-2 between Venezuela and Colombia were observed. Two cross-border transmissions were importantly noted, the first with TMRCA of approximately in 2011 and the second in 2015

## Discussion

The phylogenetic and phylogeographic analyses of DENV were performed on samples collected from patients with febrile syndrome from a northeastern Department of Colombia, which shared a border with Venezuela. This study demonstrated to favor the export and import of different strains among serotypes or clades of the same DENV serotype, whereby different DENV strains may appear by the introduction of different strains from other countries or the local evolution [26].

Globally, it was observed that the incidence rate of dengue has increased in recent decades, with southeast Asia, the western Pacific and America being the most affected regions [27]. In Colombia, DENV was first detected in the 70s and over time, because of different epidemics, four serotypes have been identified, that transformed Colombia into a hyperendemic region and made it the third leading country that reported the highest number of dengue cases in America and the Caribbean [28]. However, approximately 50% of dengue cases in the country are reported in the central eastern region of Colombia [29].

Norte de Santander is located in the northeast of Colombia, and it shares a border with Venezuela at the north and east, which is characterised by connections between the main land routes connecting Colombia and Venezuela. Currently, this area is facing difficult situations owing to crisis in Venezuela because Colombia is the main migratory destination for those intending to stay in the country, those passing through using Colombia as a bridge to reach other countries, or for pendular migration of citizens residing on the border and moving between the two countries to stock food, medicines and necessary goods that cannot be found in Venezuela. This situation has intensified since 2015 when 2,000 Colombians were deported and Migration Colombia registered the entry of 329,478 Venezuelan citizens despite closure of the border by the Venezuelan Government [30]. However, this number has been increasing recently, reaching 1,408,055 Venezuelan migrants [31]

Uncontrolled migration impacts the health of the country's citizens because the Venezuelan population is not receiving medicines and vaccines owing to shortages, causing the reappearance of infectious diseases that had been under control [32], such as diphtheria, whose first cases appeared in July 2016 and continues to be active in 2019. Similarly, measles has reappeared in Venezuela and has progressed to other countries such as Colombia, Brazil and Chile because of insufficient coverage at the borders [33]. In Venezuela, the first cases of measles were reported in July 2017, and 5,332 cases were confirmed in September 2018, including 62 deaths [34]. In Colombia, the National Institute of Health [35] confirmed 209 cases in 2018, of which 55 were imported and 117 were related to importation [36].

Despite the fact that Norte de Santander is within the region with the highest incidence rates of dengue, few epidemiological studies have been performed in the region; there are data from epidemiological surveillance conducted at the National Institute of Health and this is based on the notification of probable cases diagnosed according to the clinical symptoms of the patients. In addition, there are few laboratories in this region that perform diagnostic tests

for DENV. For example, a study was performed in the municipalities of Patios, Pamplona and Cúcuta, wherein the presence of IgG against dengue virus (DENV-IgG) was tested in 55 subjects using ELISA, finding that 84% (46/55) of the subjects had been exposed to at least one DENV serotype [37]. The relevance of this region in the country prompted the need to understand the origin and genetic diversity of DENV circulating along the northeastern border. Therefore, a phylogenetic analysis of DENV sequences collected from patients during 2015–2016 and 2018–2019 was conducted in Villa de Rosario, Norte de Santander. The comparison of genomic sequences of DENV from different strains worldwide has allowed for the establishment of the intra-serotype genetic variation and the appearance of different distinct phylogenetic clusters defined as genotypes, which have been inferred using sequence divergence of ≤6% within a region of the genome, mainly for DENV-1 and DENV-2 [38]. Thus, DENV-1 has five genotypes (genotypes I–V) and DENV-2 has six genotypes (Asian I, Asian II, Asian–American, Cosmopolitan, American and Sylvatic).

In our analyses (Tables 2 and 3), the sequence divergence within the E region was <7% inter-genotype, which is consistent with previously reported results. However, there are no criteria for the selection and name of sequence grouping below the genotype level known as lineages or clades. Therefore, this selection is based on visual segregation and branch support in the phylogenetic tree and on criteria of maximum genetic distance in some cases, whereas the name is usually designated based on its geographical origin [39] or isolation time [40]. However, there are other studies on DENV-2 that have reported inter-genotype and intra-genotype diversity of 7.3% and 2.6%, respectively [41]. In our study, the inter-genotype and intra-genotype diversity for clades and subclades was 3.3%–5.5% and 2.7%, respectively, for DENV-1 but was 2.3%–3.3% and 2.2%–3.2%, respectively, for DENV-2.

The phylogenetic analysis performed on the DENV-1 strain confirmed that it belonged to genotype V, consistent with that reported previously by Jiménez-Silva et al. [42] in samples of patients from the Ocaña region, Norte de Santander. This same genotype has also been reported by others in Santander, which is the bordering region to Norte de Santander [43–45]. Although genotype I has been reported only once in Colombia in an isolated case from Valle del Cauca in 1983 [44], genotype V is the representative genotype in most of the isolate strains in Colombia and America since its introduction in the early 70s.

In our analysis, the DENV-1 strain isolated was grouped in clade I, comprising strains from South America (SA), Central America and the Greater and Lesser Antilles. This lineage is similar to that reported by Jiménez-Silva et al., wherein the strains from Santander were grouped in a clade with sequences from Venezuela (1994–2008) and other regions of SA [42]. This has also been observed previously [46], wherein clade I comprised all sequences from Colombia and other American countries, mainly Venezuela, Argentina, Brazil, Colombia, Central America, the Caribbean and the United States (1977–2016). Moreover, in our study the inter-genotype and intra-genotype diversity for clades and subclades of DENV-1 was 3.3%–5.5% and 2.7%, respectively. However, there is no consensus on reporting this diversity for DENV-1 [41].

It is important to highlight the fact that two Colombian DENV-1 sequences obtained during 2015–2016 were grouped with a set of sequences from the southern coast of Ecuador collected during 2014 [47] and Panama (with a bootstrap value of >75%), suggesting a close genetic relationship between the viruses circulating between Colombia and Ecuador and between Colombia and Panama, further supporting the notion that the DENV flow can occur commonly among countries owing to their proximity [48–50].

In contrast, the four sequences identified in our study from the current dengue outbreak in 2019 showed a close genetic relationship to viruses from other regions of Venezuela and Colombia. Furthermore, it is important to recognize that the main serotype identified in

patients from the current DENV outbreak in Norte de Santander and other Andean regions was DENV-1 [35], whereas the entomovirological surveillance of National Institute of Health has revealed that the serotypes circulating in this region are DENV-3 and DENV-4 in *Aedes aegypti* [35].

For DENV-2, the presence of the Asian–American genotype in the sequences from Norte de Santander was corroborated, and these results were consistent with those reported in recent years in the region [9,49], and this genotype replaced the American genotype since the early 1990s [9], resulting in an increased number of serious cases in Colombia and throughout America [51]. However, it has been observed that this genotype has evolved thereafter, and evidence on the co-circulation of different clades/lineages has been reported in the country [9]. In our analysis, we found that the DENV-2 sequences were grouped in clade III, comprising sequences from SA, Central America and some countries in the Greater and Lesser Antilles, indicating that it is the most dispersed clade in America [52,53]. Moreover, we found that our sequences from 2015 and 2016 formed a paraphyletic cluster in the SA-IV subclade with other sequences from the same region and with other bordering regions with departments, such as Santander and Arauca (2013–2016), as well as with isolated strains from Venezuela during the same time period (2015) in patients from the Carabobo state [47].

Our analysis also demonstrated the circulation of two subclades (SA-IV and SA-III) comprising Colombian sequences during 2013–2016, similar to the results that were previously reported describing the circulation of lineages 1 and 2 from the Asian–American genotype of DENV-2 in Colombia and the differential geographical distributions nationwide [54]. Altogether, the above findings indicate that the circulating strains in the region belong to two different epidemiological events and provide evidence that geographical proximity causes multiple migratory events between Colombia and Venezuela. Consequently, this could explain the genetic variability of DENV, which has also been changing patterns of serotype prevalence, and the increasing incidence rate of dengue [45]. Moreover, in our study the inter-genotype and intra-genotype diversity for clades and subclades of DENV-2 was 2.3%–3.0% and 2.2%–3.2%, respectively. Other taxonomic studies have reported this diversity of 7.3% and 2.6%, respectively [41].

However, the period in which our samples were collected (2015–2016) coincided with the beginning time of the migration towards Colombia, as well as the migration of Colombians and Venezuelans to Ecuador, proximally the highest peaks for each country were during 2014 and 2017, respectively [50]. This observation is important due to the finding of with DENV-1 strains in Machala, Ecuador (2014). These exchanges have been reported in serotypes such as DENV-1, DENV-2 and DENV-3, and some researchers have found that Colombia acts as a virus export channel [42] towards Ecuador and Peru because Venezuela is probably one of the sources of virus entrance in South America [55,56]. Moreover, Colombia and Venezuela probably are single ecosystem for DENV that contributes to the dissemination of dengue across this South American region [50].

We performed robust phylogenetic and DENV evolutionary analyses using Bayesian inference methods on the envelope gene with partial sequences; this gene has been mostly used as a phylogenetic marker due to its historical, diagnostic and functional importance [41]. However, although two different RT-PCRs protocols were performed, it was not possible to amplify E gene in all DENV positive samples. This limitation may be due to the low viral load in some of the samples or to low RNA quality [57,58]. Due to the complex social situations in this border, people go to the Hospital several days after the onset of symptoms and RNA viral load is reduced when the humoral immune response appear, between 5 and 7 days after the symptomatology beginning [59], this can lead to the fragmentation of the virus genome and, therefore, to the difficulty on the obtention of the large amplicons directly from the sample. It is

important to remember that the data set that are analysed are responsible for the different estimates because the scarce or large number of sequences available from a region or country generates an inferential limitation [26,42], a greater availability of sequences is expected in these regions with advances in DENV surveillance or when next-generation sequencing becomes routine practice [41].

Finally, our study like others carried out in Asia, have shown that cross-border population behaves as a vehicle for the transmission of the DENV [60,61]. Therefore, the condition of this region as a land border between Colombia and Venezuela involves an important scenario of interest for epidemiological surveillance [62] as well as for understanding the transmission patterns and evolutionary dynamics of DENV, considering that the latest reports of the National Institute of Health in 2019 indicated that Norte de Santander was one of the regions with the highest number of dengue cases reported [33], and the fragile situation in this area is likely to complicate the whole picture. These results evidence the importance and critical roles of sentinel surveillance sites, especially along the border regions, where migrant crossing favours high transmission of infectious diseases, similar to that demonstrated with the transmission of measles from Venezuela to various countries in the regions, such as Colombia and Ecuador [34], and other arboviruses such as ZIKV [63].

## Conclusion

This study shows that DENV transmission within a high migration region as Norte de Santander enables the introduction and reintroduction of different serotypes and/or clades/lineages, leading to risks of epidemics and serious diseases caused by a new strain or new genetic variants Therefore, it is importance of conducting continuous vigilance of DENV and other arboviruses along the borders to determine the sources of viral origin and routes of propagation, that would help understand the epidemiological dynamics and preparedness for future outbreaks.

## Supporting information

**S1 Table. DENV-1 sequences used in present study.**
(XLSX)

**S2 Table. DENV-2 sequences used in present study.**
(XLSX)

## Acknowledgments

The authors would like to thank to Andrés Cardona Ríos, Dr. Javier Diaz, and Dr. Jose Usme for his advice in the phylogenetic and evolutive analyses.

## Author Contributions

**Conceptualization:** Marlen Yelitza Carrillo-Hernandez, Julian Ruiz-Saenz, Marlen Martinez-Gutierrez.

**Formal analysis:** Marlen Yelitza Carrillo-Hernandez, Julian Ruiz-Saenz.

**Funding acquisition:** Marlen Yelitza Carrillo-Hernandez, Julian Ruiz-Saenz, Marlen Martinez-Gutierrez.

**Investigation:** Marlen Yelitza Carrillo-Hernandez.

**Resources:** Lucy Jaimes-Villamizar.

**Supervision:** Sara Maria Robledo-Restrepo, Marlen Martinez-Gutierrez.

**Writing – original draft:** Marlen Yelitza Carrillo-Hernandez.

**Writing – review & editing:** Julian Ruiz-Saenz, Sara Maria Robledo-Restrepo, Marlen Martinez-Gutierrez.

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
