## [Editor Report · Decision Letter 0]

21 Sep 2020

PONE-D-20-27563

Phylogenetic and evolutionary analysis of Dengue Virus serotypes circulating at the Colombian–Venezuelan border during 2015–2019

PLOS ONE

Dear Dr. Martinez-Gutierrez,

Thank you for submitting your manuscript to PLOS ONE. After careful consideration, we feel that it has merit but does not fully meet PLOS ONE’s publication criteria as it currently stands. Therefore, we invite you to submit a revised version of the manuscript that addresses the points raised during the review process.

Despite the good intentions of your group in obtaining a certification of readability, I regret to inform you that the job of editing was not done satisfactorily.  I find that the errors are so numerous that I cannot in good conscience invite reviewers to wade through your manuscript.  Indeed, even I have not read the whole thing. So far, I have noticed the following

l. 99 spelling of "April"

l. 97 since 'eastern" is an adjective, a noun needs to follow.

l. 98  Villa Rosario is at (not "has")

l 101 personnel, not "personal"

l 104 them  they

l. 162 was and exponential ???

These examples were enough to notify me that this manuscript is not acceptable as it now stands. I did get the impression that, if extensively rewritten, there may be information worthy of publication. Certainly, any resubmission will need thorough peer review.

We look forward to receiving your revised manuscript.

Kind regards,

Ulrich Melcher

Academic Editor

PLOS ONE

Journal Requirements:

3.We note that [Figure(s) 1] in your submission contain [map/satellite] images which may be copyrighted. All PLOS content is published under the Creative Commons Attribution License (CC BY 4.0), which means that the manuscript, images, and Supporting Information files will be freely available online, and any third party is permitted to access, download, copy, distribute, and use these materials in any way, even commercially, with proper attribution. For these reasons, we cannot publish previously copyrighted maps or satellite images created using proprietary data, such as Google software (Google Maps, Street View, and Earth). For more information, see our copyright guidelines: http://journals.plos.org/plosone/s/licenses-and-copyright.

1.    You may seek permission from the original copyright holder of Figure(s) [1] to publish the content specifically under the CC BY 4.0 license. 

Additional Editor Comments (if provided):

Despite the good intentions of your group in obtaining a certification of readability, I regret to inform you that the job of editing was not done satisfactorily. I find that the errors are so numerous that I cannot in good conscience invite reviewers to wade through your manuscript. Indeed, even I have not read the whole thing. So far, I have notice the following

l. 99 spelling of "April"

l. 97 since 'eastern" is an adjective, a noun needs to follow.

l. 98 Villa Rosario is at (not "has")

l 101 personnel, not "personal"

l 104 them  they

l. 162 was and exponential ???

These examples were enough to notfy me that this mansucript is not acceptable as it now stands. I did get the impression that, if extensively rewritten, there may be information worthy of publication. Certainly, any resubmission will need thorough peer review.
---

## [Author Response · Author response to Decision Letter 0]

4 Nov 2020

According to your suggestions we requested a second revision of the English by ENAGO (the editing brand of Crimson Interactive Inc. under Translation + Editing) of our original paper Phylogenetic and evolutive analysis of Dengue Virus strains circulating at the Colombian–Venezuelan border during 2015–2019. 

We have uploaded a marked-up copy of our manuscript that highlights changes made to the original version (labeled *Revised Manuscript with Track Changes*) and a second file with our revised paper without tracked changes (labeled *Manuscript*). Moreover, we attach to this letter the new certificate from ENAGO.

In the case of figure 1 that contains a fragment of the satellite map, we decided to eliminate it.

---

## [Decision Letter · Decision Letter 1]

22 Mar 2021

PONE-D-20-27563R1

Phylogenetic and evolutionary analysis of Dengue Virus serotypes circulating at the Colombian–Venezuelan border during 2015–2019

PLOS ONE

Dear Dr. Martinez-Gutierrez,

Thank you for submitting your manuscript to PLOS ONE. After careful consideration, we feel that it has merit but does not fully meet PLOS ONE’s publication criteria as it currently stands. Therefore, we invite you to submit a revised version of the manuscript that addresses the points raised during the review process.

In this study, the authors collected the clinical samples during 2015-2016 and 2018-2019. The use of 2015–2019 in the Title was inappropriate.The sample size of 229 suspected DENV samples was relatively small. In particular, of 41 samples positive for DENV, only 10 E gene sequences were obtained. Why so few sequences to be obtained? The proportion of successfully sequencing was substantially lower than a previous report (J Travel Med. 2020, 27(7): taaa195). Could the authors use RT-nested PCR to amplify the E gene (Emerg Infect Dis. 2012, 18(11):1850-7) ?  Too few sequences do not result in a solid result or conclusion. This point should be mentioned as a limitation.The molecular epidemiology of four DENV serotypes in America (especially in Venezuela and Colombia) should be mentioned in Introduction. If four DENV serotypes are co-circulating in a continent/region, cross-border transmission of all four serotypes were often observed among neighboring countries. The compelling evidences are from Asia, where all four DENV serotypes were introduced into China from neighboring countries along land border ports (e.g. China-Myanmar border), and/or surrounding countries via air travel (e.g. between Shanghai, China and some Southeastern Asian countries) (Emerg Infect Dis 2018; 24:1756–8，J Travel Med. 2020, 27: taaa195). If all four serotypes co-circulating in America, why no DENV-3 and DENV-4 to be detected? And also no cross-border transmission of DENV-3 and DENV-4 in these regions? This point should be mentioned and discussed (maybe as a limitation) in the paper. the cross-border transmission of all four serotypes were observed betweenThe structure of the Results should be reorganized. Too many sections are included in Results. Phylogenetic analysis, Phylogenetic analysis of DENV-1, and Phylogenetic analysis of DENV-2 should be merged into one section (e.g. Phylogenetic analysis). Similarly, Evolutionary analysis, DENV-1 E gene phylogeography and DENV-2 E gene phylogeography should be merged into one section of “Phylogeographic analyses”.In the Phylogenetic analyses of DENV-1 and -2, it makes no sense to further divide genotypes into clades and even sub-clades. I suggest to remove these classification.In the phylogeographic analysis of DENV-1, first introduction should not be traced back to 1973 (it should be later than 1973, maybe about 1980). In addition, the identification of second, third and fourth introductions was incorrect. The so-called three introduction events were more likely to be a single introduction event with a tMRCA in around 1994 (at the next node of the 1993 node).In the phylogeographic analysis of DENV-2, there might be two independent introduction events of DENV-2 from Venezuela to Colombia in around 1998 and 2001. Since then, frequent cross-border transmissions of DENV-2 between Venezuela and Colombia were observed. So, the description of the so-called fifth and sixth introduction events were inappropriate.  In the phylogeographic analyses, the sequences from other continents (e.g. Asia, Africa) should be included.

We look forward to receiving your revised manuscript.

Kind regards,

Chiyu Zhang, Ph.D.

Academic Editor

PLOS ONE

Additional Editor Comments (if provided):

1. In this study, the authors collected the clinical samples during 2015-2016 and 2018-2019. The use of 2015–2019 in the Title was inappropriate.

2. The sample size of 229 suspected DENV samples was relatively small. In particular, of 41 samples positive for DENV, only 10 E gene sequences were obtained. Why so few sequences to be obtained? The proportion of successfully sequencing was substantially lower than a previous report (J Travel Med. 2020, 27(7): taaa195). Could the authors use RT-nested PCR to amplify the E gene (Emerg Infect Dis. 2012, 18(11):1850-7) ? Too few sequences do not result in a solid result or conclusion. This point should be mentioned as a limitation.

3. The molecular epidemiology of four DENV serotypes in America (especially in Venezuela and Colombia) should be mentioned in Introduction. If four DENV serotypes are co-circulating in a continent/region, cross-border transmission of all four serotypes were often observed among neighboring countries. The compelling evidences are from Asia, where all four DENV serotypes were introduced into China from neighboring countries along land border ports (e.g. China-Myanmar border), and/or surrounding countries via air travel (e.g. between Shanghai, China and some Southeastern Asian countries) (Emerg Infect Dis 2018; 24:1756–8，J Travel Med. 2020, 27: taaa195). If all four serotypes co-circulating in America, why no DENV-3 and DENV-4 to be detected? And also no cross-border transmission of DENV-3 and DENV-4 in these regions? This point should be mentioned and discussed (maybe as a limitation) in the paper.

4. the cross-border transmission of all four serotypes were observed between

5. The structure of the Results should be reorganized. Too many sections are included in Results. Phylogenetic analysis, Phylogenetic analysis of DENV-1, and Phylogenetic analysis of DENV-2 should be merged into one section (e.g. Phylogenetic analysis). Similarly, Evolutionary analysis, DENV-1 E gene phylogeography and DENV-2 E gene phylogeography should be merged into one section of “Phylogeographic analyses”.

6. In the Phylogenetic analyses of DENV-1 and -2, it makes no sense to further divide genotypes into clades and even sub-clades. I suggest to remove these classification.

7. In the phylogeographic analysis of DENV-1, first introduction should not be traced back to 1973 (it should be later than 1973, maybe about 1980). In addition, the identification of second, third and fourth introductions was incorrect. The so-called three introduction events were more likely to be a single introduction event with a tMRCA in around 1994 (at the next node of the 1993 node).

8. In the phylogeographic analysis of DENV-2, there might be two independent introduction events of DENV-2 from Venezuela to Colombia in around 1998 and 2001. Since then, frequent cross-border transmissions of DENV-2 between Venezuela and Colombia were observed. So, the description of the so-called fifth and sixth introduction events were inappropriate.

9. In the phylogeographic analyses, the sequences from other continents (e.g. Asia, Africa) should be included.

Reviewers' comments:

Reviewer's Responses to Questions

**Comments to the Author**

1. If the authors have adequately addressed your comments raised in a previous round of review and you feel that this manuscript is now acceptable for publication, you may indicate that here to bypass the “Comments to the Author” section, enter your conflict of interest statement in the “Confidential to Editor” section, and submit your "Accept" recommendation.

Reviewer #1: (No Response)

Reviewer #2: All comments have been addressed

2. Is the manuscript technically sound, and do the data support the conclusions?

Reviewer #1: Yes

Reviewer #2: Yes

3. Has the statistical analysis been performed appropriately and rigorously? 

Reviewer #1: Yes

Reviewer #2: Yes

4. Have the authors made all data underlying the findings in their manuscript fully available?

Reviewer #1: Yes

Reviewer #2: Yes

5. Is the manuscript presented in an intelligible fashion and written in standard English?

Reviewer #1: No

Reviewer #2: Yes

6. Review Comments to the Author

Reviewer #1: In this paper authors track the migration of the dengue strain DENV in Central America, especially in the Columbia – Venezuela border. The paper is well-written. I am not expert in the phylogenetics approaches, but with whatever knowledge I have, I think methods are well done. Results are presented well, but I did not understand, why there is suddenly figure legend in the result text, I hope, this legend will go below the figure in the final version.

Also, I have suggestion to give. In the discussion, I was lost in reading how the dengue strain travelled through Columbia, Venezuela and other parts of Central America. I was thinking if this information can be depicted through a map, it will be beneficial to readers. Specially for people who are not familiar to the Central America geography.

Reviewer #2: The authors conduct a phylogenetic analysis of DENV strains circulating in the border of

Colombia and Venezuela. The results of this study provide some support for DENV cross-border transmission and suggests that border surveillance and characterization of imported and exported strains are very important. However, this study has many limitations.

1. The number of sequences included in the study was too small to illuminate the genetic characteristics of the border-prevalent strains. So the conclusions, geographical proximity between Colombia and Venezuela is favourable for the export and import of different cannot be strongly supported.

2. Is there more Asian strains included for the evolutionary rate calculating? Too few sequences will lead to bias in the calculation results. There are a large number of dengue sequences from Asia in the database.

3. It will be better to annotate the meanings of the different colors in Figure 2 and Figure 3 in the figure.

7. PLOS authors have the option to publish the peer review history of their article (what does this mean?). If published, this will include your full peer review and any attached files.

Reviewer #1: No

Reviewer #2: No

---

## [Author Response · Author response to Decision Letter 1]

5 May 2021

Additional Editor Comments

1. In this study, the authors collected the clinical samples during 2015-2016 and 2018-2019. The use of 2015–2019 in the Title was inappropriate.

Answer from author:

We agree with the editor comment and modify the title according to the recommendation (Lines 1-2)

2. The sample size of 229 suspected DENV samples was relatively small. In particular, of 41 samples positive for DENV, only 10 E gene sequences were obtained. Why so few sequences to be obtained? The proportion of successfully sequencing was substantially lower than a previous report (J Travel Med. 2020, 27(7): taaa195). Could the authors use RT-nested PCR to amplify the E gene (Emerg Infect Dis. 2012, 18(11):1850-7) ? Too few sequences do not result in a solid result or conclusion. This point should be mentioned as a limitation.

Answer from author:

We agree to the reviewer; the low number could be a Limitation. However, it is clear that our research area has a different epidemiological panorama in comparison to the one described by Ma et al., in Shanghai (J Travel Med. 2020, 27(7): taaa195). In the paper mentioned by the reviewer, most of the DENV cases were imported from Southeast Asian countries (aprox 60%). In the studied Colombian–Venezuelan border we have published that although patients usually are described as DENV, most of the cases could be other viruses such as Chikungunya, Zika or other unidentified pathogens (Carrillo et al., doi: 10.1186/s12879-018-2976-1). Therefore, to clarify this point, a short sentence about this topic was added to the Discussion (line 501-504). Besides, One Step-rtPCR was used to obtain ENV sequences in low RNA quality and problematic Samples. This was also better explained in the Methods Section (Line 138-144). 

3. The molecular epidemiology of four DENV serotypes in America (especially in Venezuela and Colombia) should be mentioned in Introduction. If four DENV serotypes are co-circulating in a continent/region, cross-border transmission of all four serotypes were often observed among neighboring countries. The compelling evidences are from Asia, where all four DENV serotypes were introduced into China from neighboring countries along land border ports (e.g. China-Myanmar border), and/or surrounding countries via air travel (e.g. between Shanghai, China and some Southeastern Asian countries) (Emerg Infect Dis 2018; 24:1756–8，J Travel Med. 2020, 27: taaa195). If all four serotypes co-circulating in America, why no DENV-3 and DENV-4 to be detected? And also, no cross-border transmission of DENV-3 and DENV-4 in these regions? This point should be mentioned and discussed (maybe as a limitation) in the paper.

Answer from author:

We thank the editor for the recommendation and included a short paragraph in the Introduction to explain the molecular epidemiology of DENV in Venezuela (Line 67-71). Also, we modified the Discussion to clarifies the information (Line 504-508).

3. the cross-border transmission of all four serotypes were observed between

Answer from author:

Unfortunately, we do not understand the editor comment.

4. The structure of the Results should be reorganized. Too many sections are included in Results. Phylogenetic analysis, Phylogenetic analysis of DENV-1, and Phylogenetic analysis of DENV-2 should be merged into one section (e.g. Phylogenetic analysis). Similarly, Evolutionary analysis, DENV-1 E gene phylogeography and DENV-2 E gene phylogeography should be merged into one section of “Phylogeographic analyses”.

Answer from author:

We agree to the editor. The results were modified according to the recommendation to a better understanding of the information.

5. In the Phylogenetic analyses of DENV-1 and -2, it makes no sense to further divide genotypes into clades and even sub-clades. I suggest to remove these classification.

Answer from author:

We partially disagree to the reviewer. Although it has been widely discussed the use of Genotypes subclassifications into Clades and subclades; in our paper, this classification allows us to understand inter-genotype and intra-genotype viral genome diversity. Besides, it allows to achieve a full comparison to other previously published papers in the region of the Americas. We support the use of this subclassification with some references in the Discussion section (lines 422-428)

6. In the phylogeographic analysis of DENV-1, first introduction should not be traced back to 1973 (it should be later than 1973, maybe about 1980). In addition, the identification of second, third and fourth introductions was incorrect. The so-called three introduction events were more likely to be a single introduction event with a tMRCA in around 1994 (at the next node of the 1993 node).

Answer from author:

We agree to the reviewer and apologize for the involuntary mistake in the presented data for DENV-1. We developed a new phylogeographic analysis for DENV-1 by including missing sequences. The full data and trees fully agree to the reviewer corrections and were included in the new version of the manuscript (Lines 285-288 and 290-320) and a new figure 4 was included too. 

7. In the phylogeographic analysis of DENV-2, there might be two independent introduction events of DENV-2 from Venezuela to Colombia in around 1998 and 2001. Since then, frequent cross-border transmissions of DENV-2 between Venezuela and Colombia were observed. So, the description of the so-called fifth and sixth introduction events were inappropriate.

Answer from author:

We fully agree to the editor. The data analysis was subject to edition according to the recommendation and the corrections were included in the new version of the manuscript (Lines 285-288 and 322-362) and in the new figure 5 too. 

8. In the phylogeographic analyses, the sequences from other continents (e.g. Asia, Africa) should be included.

Answer from author:

We agree to the reviewer. New sequences from other continents were added, the results were changed (Lines 285-362) to the dataset corresponding to Figure 4 and 5.

Reviewer 1: 

In this paper authors track the migration of the dengue strain DENV in Central America, especially in the Columbia – Venezuela border. The paper is well-written. I am not expert in the phylogenetics approaches, but with whatever knowledge I have, I think methods are well done. Results are presented well, but I did not understand, why there is suddenly figure legend in the result text, I hope, this legend will go below the figure in the final version.

Answer from author:

We thank the reviewer for him/her words and agree to the last comment. However, this is part of the PLOS Template.

Also, I have suggestion to give. In the discussion, I was lost in reading how the dengue strain travelled through Columbia, Venezuela and other parts of Central America. I was thinking if this information can be depicted through a map, it will be beneficial to readers. Specially for people who are not familiar to the Central America geography.

Answer from author:

We agree to the reviewer. According to this recommendation, we added a new map in the part A of figure 1 (Lines 199-201). 

Reviewer 2: 

The authors conduct a phylogenetic analysis of DENV strains circulating in the border of Colombia and Venezuela. The results of this study provide some support for DENV cross-border transmission and suggests that border surveillance and characterization of imported and exported strains are very important. However, this study has many limitations. 

1. The number of sequences included in the study was too small to illuminate the genetic characteristics of the border-prevalent strains. So the conclusions, geographical proximity between Colombia and Venezuela is favourable for the export and import of different cannot be strongly supported.

Answer from author:

We partially agree to the reviewer. We discuss the size of the sample as a limitation; however, the inclusion of several sequences previously reported in this country’s trough the las years, strongly support the conclusions. We included some sentences in the Discussion to clarify this item (Lines 501-504).

2. Is there more Asian strains included for the evolutionary rate calculating? Too few sequences will lead to bias in the calculation results. There are a large number of dengue sequences from Asia in the database.

Answer from author:

We agree to the reviewer. We included new sequences from different continents into the dataset to clarify the results (Lines 285-288) and the figures 4 and 5 were changed. 

3. It will be better to annotate the meanings of the different colors in Figure 2 and Figure 3.

Answer from author:

We totally agree to the reviewer. The figure legends (Figure 2, lines 237-239 and Figure 3, lines 264-265) where modified according to the recommendation.

---

## [Editor Report · Decision Letter 2]

9 May 2021

PONE-D-20-27563R2

Phylogenetic and evolutionary analysis of dengue virus serotypes circulating at the Colombian–Venezuelan border during 2015-2016 and 2018-2019

PLOS ONE

Dear Dr. Martinez-Gutierrez,

Thank you for submitting your manuscript to PLOS ONE. After careful consideration, we feel that it has merit but does not fully meet PLOS ONE’s publication criteria as it currently stands. Therefore, we invite you to submit a revised version of the manuscript that addresses the points raised during the review process.

Please carefully check the Figures 4 and 5, and see my new comments. 

We look forward to receiving your revised manuscript.

Kind regards,

Chiyu Zhang, Ph.D.

Academic Editor

PLOS ONE

Journal Requirements:

Additional Editor Comments (if provided):

1. Please carefully check the Figures 4 and 5. Do the authors sure that the sampling year of the latest samples (the terminal braches in the MCC trees) were about 2025? Therefore, all tMRCA of the key nodes were wrong. In addition, please carefully determine which nodes can reflect the tMRCA of the strains from Colombia.

2. Please clearly label each introduction event (at corresponding node).

3. Line141：previously reported primers: please provide refs.

4. Line 315: sequences from Santander reported during 2010–2016) and Venezuela (2005–2015): please check this sentence.

---

## [Author Response · Author response to Decision Letter 2]

13 May 2021

1. Please carefully check the Figures 4 and 5. Do the authors sure that the sampling year of the latest samples (the terminal braches in the MCC trees) were about 2025? Therefore, all tMRCA of the key nodes were wrong. In addition, please carefully determine which nodes can reflect the tMRCA of the strains from Colombia.

Answer from author:

We apologies for the mistakes in the figures. According to the revision we corrected the terminal branches un the MCC trees (Figure 4 and 5). Moreover, all tMRCA of the key nodes were corrected. Finally, all the corrections were included in the new version of the manuscript (Lines 311, 312, 320, 325, 326, 344, 345, 349, 351, 352, 357, 358 and 362 

2. Please clearly label each introduction event (at corresponding node).

Answer from author:

We checked and corrected each introduction event in the new figures 4 and 5.

3. Line141：previously reported primers: please provide refs.

Answer from author:

We provided the reference to the end of the sentence (Reference number 19). 

4. Line 315: sequences from Santander reported during 2010–2016) and Venezuela (2005–2015): please check this sentence.

Answer from author:

The sentence was corrected in the lines 314 to 316.

---

## [Editor Report · Decision Letter 3]

17 May 2021

Phylogenetic and evolutionary analysis of dengue virus serotypes circulating at the Colombian–Venezuelan border during 2015-2016 and 2018-2019

PONE-D-20-27563R3

Dear Dr. Martinez-Gutierrez,

We’re pleased to inform you that your manuscript has been judged scientifically suitable for publication and will be formally accepted for publication once it meets all outstanding technical requirements.

Kind regards,

Chiyu Zhang, Ph.D.

Academic Editor

PLOS ONE

Additional Editor Comments (optional):

line351:"(95% HPD: 1995–1995)" should be " (95% HPD: 1992–1995)".
---

## [Editor Report · Acceptance letter]

21 May 2021

PONE-D-20-27563R3 

Phylogenetic and evolutionary analysis of dengue virus serotypes circulating at the Colombian–Venezuelan border during 2015-2016 and 2018-2019 

Dear Dr. Martinez-Gutierrez:

I'm pleased to inform you that your manuscript has been deemed suitable for publication in PLOS ONE. Congratulations! Your manuscript is now with our production department. 

Kind regards, 

on behalf of

Dr. Chiyu Zhang 

Academic Editor

PLOS ONE